

# Sulforaphene suppressed cell proliferation and promoted apoptosis of COV362 cells in endometrioid ovarian cancer

Hui-Yan Yu[1], Li Yang[1], Yuan-Cai Liu[2] and Ai-Jun Yu[1]

[1] Zhejiang Cancer Hospital, Zhejiang, China
[2] Zhejiang Chinese Medical University, Zhejiang, China

## ABSTRACT

**Aim**. N6-methyladenosine (m6A) RNA methylation exerts a regulatory effect on endometrioid ovarian cancer (EOC), but the specific m6A regulator genes in EOC remain to be explored. This study investigated that sulforaphene (Sul) is implicated in EOC development by regulating methyltransferase-like 3 (METTL3).

**Methods**. The dysregulated m6A RNA methylation genes in EOC were determined by methylated RNA immunoprecipitation (MeRIP-seq) and RNA sequencing. The roles of METTL3 and/or Sul on viability, proliferative ability, cell cycle, and apoptosis of EOC cells were determined by MTT, colony formation, flow cytometry, and TUNEL staining assay, respectively. The expression of METTL3 and apoptosis-related proteins in EOC cells was detected by quantitative real-time polymerase chain reaction (qRT-PCR) and western blot assays.

**Results**. Five m6A RNA methylation regulators (METTL3, ELF3, IGF2BP2, FTO, and METTL14) were differentially expressed in EOC, among which METTL3 had the highest expression level. Silencing METTL3 reduced the clonal expansion and viability of EOC cells, and caused the cells to arrest in the G0/G1 phase. This also promoted apoptosis in the EOC cells and activated the FAS/FADD and mitochondrial apoptosis pathways. In contrast, overexpressing METTL3 had the opposite effect. Sul, in a dose-dependent manner, reduced the viability of EOC cells but promoted their apoptosis. Sul also increased the levels of IGF2BP2 and FAS, while decreasing the levels of KRT8 and METTL3. Furthermore, Sul was able to reverse the effects of METTL3 overexpression on EOC cells.

**Conclusions**. Sul could suppress cell proliferation and promote apoptosis of EOC cells by inhibiting the METTL3 to activate the FAS/FADD and apoptosis-associated pathways.

Corresponding author
Ai-Jun Yu, Yaj1993@126.com

## INTRODUCTION

About one-tenth of all epithelial ovarian cancers are endometrioid ovarian cancers (EOC), a rare kind of ovarian cancer (OC) (*Pierson et al., 2020*). The development of EOC is related to endometriosis. Approximately 26% of EOC patients were found to have endometriosis when undergoing surgery, while the probability of endometriosis in the uterus of other tumor patients was less than 6% (*Wilbur et al., 2017*). Patients with recurrent EOC have a

dismal prognosis, although the fact that EOC has a better prognosis than other kinds of OC (*Pujade-Lauraine & Combe, 2016*). To further enhance the prognosis of patients with EOC, it is vital to discover safe and efficient medications.

Inducing apoptosis is an important way to treat malignant tumors, and it has also become one of the important strategies to treat EOC. As a classical apoptotic mechanism, the FAS/FAS ligand (FASL) pathway has emerged as the primary method of cancer treatment (*Peng et al., 2022*). It is worth noting that the genetic polymorphism of FAS and FASL, as well as the low or nonexistent expression of the FAS gene in EOC, are coupled to the risk and prognosis of patients with EOC (*Li et al., 2013*; *Chaudhry, Srinivasan & Patel, 2010*), which means that focusing on FAS/FASL pathway may be a crucial way to treat EOC. Interestingly, N6-methyl adenine (M6A), a reversible mRNA modification method, has been reported to exert a vital effect on many pathways such as mammalian apoptosis and tumorigenesis (*Sun, Wu & Ming, 2019*). The methylation process of m6A RNA is mainly controlled by m6A methyltransferase, m6A demethylase, and m6A reading protein. Methyltransferase-like 3 (METTL3) is an m6A methyltransferase, which is the catalytic core of the m6A complex (*Gao et al., 2021*). A study by *Bi et al. (2021a)*; *Bi et al. (2021b)* pointed out that METTL3 was over-expressed in OC, resulting in a decrease in the apoptosis rate of OC, which may be attributed to the METTL3-mediated increase of m6A methylation of mRNA related to apoptosis (*Xu et al., 2021*). Therefore, targeting METTL3 may regulate the alterations of the apoptosis signaling pathway downstream of FAS by mediating the methylation of mRNA.

The active component in Cruciferae plants known as sulforaphene (Sul) has been shown to have numerous anticancer properties, including the ability to induce tumor cell apoptosis and suppress tumor cell proliferation (*Chatterjee, Rhee & Ahn, 2016*; *Pawlik et al., 2017*). Additionally, Sul has an anti-cancer function in OC, and the underlying mechanism is connected to the control of the cell cycle and apoptosis (*Hudecova et al., 2016*; *Chang et al., 2013*). However, uncertainty exists over whether Sul controls the growth of EOC by a comparable mechanism. Notably, *Lewinska et al. (2017)* pointed out that the anticancer effect of Sul in breast cancer was associated with the regulation of m6A RNA methylation. Nevertheless, the impact of Sul on methylation regulation of m6A RNA has not been proved in EOC. Therefore, the purpose of this investigation is to establish if Sul can perform an anti-tumor effect by controlling the methylation of mRNA, which is mediated by METTL3.

## MATERIALS AND METHODS

### Cell culture
Human EOC cell line COV362 was ordered from Nanjing Cobioer Biosciences Co. Ltd (Nanjing, China) and was cultured with DMEM complete medium (CBP50008, Cobioer, China) at 37 °C incubator with 5% $CO_2$.

### Clinical specimens
EOC samples ($n = 3$) and ovarian endometriosis (OE) samples ($n = 3$) were obtained from patients who underwent surgery at Zhejiang Cancer Hospital and were histopathologically

assessed by two trained gynecological pathologists before use in this investigation. Forms for informed consent were signed by all patients. The Zhejiang Cancer Hospital Ethics Committee gave its approval for the collection and analysis of human samples (No. IRB-2021-350).

## Methylated RNA immunoprecipitation (Me-RIP)-sequencing and data analysis

The procedure of sequencing was conducted as previously described (*Yang et al., 2022*). Total RNA from COV362 cells was extracted using Trizol (T11196, Saint-Bio, China), the purity and integrity of which were determined by Qubit 4 (Thermo Fisher, Waltham, MA, USA) and Bioanalyzer 2100 (Agilent, Santa Clara, CA, USA). RNA was captured by Dynabeads Oligo (dT) (25-61005, Thermo Fisher, Scientific, Waltham, MA, USA), and the obtained RNA was broken into small fragments of about 100 nt and immunoprecipitated with m6A-specific antibodies (202003, Synaptic Systems, Goettingen, Germany). The collected RNA fragments containing m6A modifications were used for the construction of a cDNA library using the dUTP method. Finally, double-ended sequencing was conducted using Illumina Novaseq™ 6000 according to standard procedures. The samples' quality was checked by Fastp software, after which HISAT2 mapped the data onto the genome. With visualization provided by the IGV (*Thorvaldsdóttir, Robinson & Mesirov, 2013*) program and annotation provided by ChIPseeker (*Yu, Wang & He, 2015*), ExomePeak (*Meng et al., 2014*) was used for peak calling analysis and gene difference peak analysis. StringTie (*Pertea et al., 2015*) was used for quantification, HOMER and MEME2 were utilized for motif analysis, and edgeR (*Robinson, McCarthy & Smyth, 2010*) was used for difference analysis.

## Cell transfection

Specific small interfering RNA (siRNA) targeting METTL3 (si-METTL3, si-METTL3# 1:AGCTACAGATCCTGAGTTAGAGA,si-METTL3#2:GAGTTGATTGAGGTAAAGCGAGG, si-METTL3#3: ATGTTGATCTGGAGATAGAGAGC) and its negative control (si-NC) were synthesized by Genscript (Jiangsu, China). The overexpression vector for METTL3 was constructed by cloning its full-length sequence into the pcDNA3.1 vector (pcDNA3.1-METTL3, VT1001, Youbio, China). To knock down or over-express METTL3 in cells, the synthesized sequences or plasmids were transfected into COV362 cells by different transfection reagents (11668027 and 13778030, Thermo Fisher, Waltham, MA, USA). Transfection reagents and plasmids or siRNA were diluted in advance with serum-free medium and subsequently mixed and incubated with cells for 48 h.

## Cell treatment

The experiment was conducted in three stages. First, the effects of overexpression or silencing of METTL3 on EOC cells were determined by the loss and gain of function experiment. Then, the cells were incubated with different concentrations of sulforaphene (Sul, 20, 40, 60 µM, HY-N2450, MedChemExpress, Monmouth Junction, NJ, China) to assess the effect of Sul. Finally, the cells overexpressing METTL3 were incubated with Sul (60 µM) to assess whether Sul could block the effect of pcDNA3.1-METTL3.

## Quantitative real-time polymerase chain reaction (QRT-PCR)

Total RNA from COV362 cells was extracted using Trizol reagent (Thermo Fisher Scientific, Waltham, MA, USA). A Nanodrop 2000 instrument (Thermo Fisher Scientific, Waltham, MA, USA) was used to measure RNA concentrations. The cDNA was generated from RNA using an RT reagent Kit (CW2569, CWBIO, Beijing, China). Thereafter, qPCR was performed using qPCR SuperMix (AQ131, TransGen Biotech, Beijing, China). It was performed using a CFX Connect system (Bio-rad, USA). at 95 °C for 600 s, 95 °C for 10 s, 65 °C for 60 s, 97 °C for 1 s, and 37 °C for 30 s, for a total of 40 cycles. The primer sequences for RT-PCR are displayed as follows: METTL3, forward: 5′-CCCTATGGGACCCTGACAG-3′, reverse: 5′-CTGGTTGAAGCCTTGGGGAT-3′; ELF3, forward: 5′-CCACTCCGGTAGCCTCATGG-3′, reverse: 5′-AAACCATCGCTGGGGAAGAG-3′; IGF2BP2, forward: 5′-ACTGCAGGCTAAGGGAGAGA-3′, reverse: 5′-CGCAGCGGGAAATCAATCTG-3′; FTO, forward: 5′-TGATCTCAATGCCACCCACC-3′, reverse: 5′-TGTGCCTTATCAAC-CTGGGAG-3′; METTL14, forward: 5′-GTAGCACAGACGGG-GACTTC-3′, reverse: 5′-GCCAGCCTGGTCGAATTGTA-3′; Fas, forward: 5′-CGGAGTTGGGGAAGCTCTTT-3′, reverse: 5′-TTTGGTGCAAGGGTCACAGT-3′; KRT8, forward: 5′-GAATGAATGGGGT-GAGCTGGA-3′, reverse: 5′-TCTGGTTGACCGTAACTGCG-3′; GAPDH, forward: 5′-CGGAGTCAACGGATTTGGTCGTAT-3′, reverse: 5′-AGCCTTCTCCATGGTGGTGAAGAC-3′, GAPDH was regarded as the internal reference. The $2^{-\Delta\Delta CT}$ method was used for calculating the relative transcription level of the target gene.

## Western blot assay

COV362 cells were firstly lysed in RIPA Buffer (T10272, Saint-Bio, China) and centrifuged at $12,000\times$ g for 10 min to obtain the supernatant, the concentration of which were evaluated by BCA Assay Kit (34001, Saint-Bio, China). Later, a 10% sodium dodecyl sulfate-polyacrylamide gel electrophoresis (SDS-PAGE) gel was employed for the separation of proteins that were electro-transferred onto a membrane (IPVH00010, Saint-Bio, China). After the non-specific sites were blocked by 5% (w/v) nonfat dry milk, the membrane was reacted with primary antibodies against METTL3 (1:1000, ab195352, Abcam, Boston, MA, USA), FAS (1:1000, AF5342, Affinity, San Francisco, CA, USA), phospho-FADD (1:1000, DF2996, Affinity), FADD (1:1000, DF7674, Affinity), Caspase-8 (1:1000, AF6442, Affinity, San Francisco, CA, USA), Caspase-3 (1:1000, AF6311, Affinity, San Francisco, CA, USA), Bcl-2 (1:1000, AF6139, Affinity, San Francisco, CA, USA), Bax (1:1000, AF0120, Affinity, San Francisco, CA, USA), and GAPDH (1:5000, AF7021, Affinity, San Francisco, CA, USA) overnight at 4 °C. GAPDH was regarded as the internal reference. Following the incubation with secondary HRP-linked anti-rabbit IgG antibody (1:3000, #7074, Cell Signaling Technology, Danvers, MA, USA) for 2 h, the band signals were detected by a Chemiluminescence imaging system (ChemiScope 6000, Clinex, China) equipped with ECL reagent kits (T15139, Saint-Bio, China). Finally, the band intensity was measured by Image J software (Bethesda, MD, USA).

## MTT assay

MTT kit (20311, Saint-Bio, China) was employed to determine the cell viability. Briefly, cells were planted into 96-well plates and then treated differently according to the experimental requirements. 20 µL MTT was added to the reaction well and incubated for 4 h. After the purple crystal was dissolved by the Formazan reagent, the absorbance (570 nm) of the reaction well was determined using a microplate plate reader (SpectraMax Mini, Molecular Devices, San Jose, CA, USA). The half maximal inhibitory concentration (IC50) of Sul was calculated according to the results.

## Colony formation assay

The treated cells were first seeded into a 6-well plate with 1,000 cells per well and cultured there for 14 days with a medium change every three days. After the culture, the cells were fixed with paraformaldehyde solution (R23025, Saint-Bio, China) and stained with crystal violet (R21877, Saint-Bio, China), after which the number of colonies was counted under a microscope (ECLIPSE E100, Nikon, Tokyo, Japan).

## Cell cycle and apoptosis assay

The cell cycle and apoptotic cells were determined by the Cell cycle kit (R21806, Saint-Bio, China) and Annexin V-FITC/ Propidium Iodide (PI) double staining kit (G003, Nanjing Jiancheng Bioengineering Institute, Nanjing, China), respectively. For cell cycle examination, cells were centrifuged at $1,000\times$ g for 5 min and then re-suspended in pre-cooled phosphate buffer saline (PBS). After centrifugation in the same way, the cells were resuspended in precooled 70% ethanol and kept for 2 h. Then, after washing the cells with PBS, the cells were stained with a PI solution for 30 min. The results were analyzed by flow cytometry (CytoFLEX LX, Beckman Coulter, Brea, CA, USA). For cell apoptosis examination, cells were centrifuged at $1,000\times$ g for 5 min and then re-suspended in binding buffer, after which the cells were stained with Annexin V-FITC solution (5 µL) and PI solution (5 µL) for 10 min. The results were analyzed by flow cytometry.

## Terminal deoxynucleotidyl transferase dUTP nick end labeling assay

The Terminal deoxynucleotidyl transferase dUTP nick end labeling (TUNEL) (red) Cell Apoptosis Detection Kit (G1502, Servicebio, Wuhan, China) was employed to detect the apoptotic cells. Cells were inoculated on chamber slides and then treated differently according to the experimental requirements. The cells were fixed for 20 min by adding paraformaldehyde solution to each chamber slide, after which cells were immersed in Triton-X-100 solution (T16608, Saint-Bio, China) for incubation for 5 min. After washing the sample with PBS twice, the liquid on the chamber slides was absorbed with filter paper. Each sample was dripped with 50 µL of equalization buffer and incubated for 10 min. After washing, the cells were incubated with TdT incubation buffer for 1 h and then DAPI solution (G1012, Servicebio, Wuhan, China) for 8 min. Following dyeing, an anti-fade mounting medium (G1401, Servicebio, Wuhan, China) was used to seal the samples, and the samples were analyzed under a fluorescence microscope (Nikon Eclipse C1, Nikon, Tokyo, Japan).

## Statistical analysis

All cell experiments were repeated three times and statistical analysis was realized by adopting SPSS 16.0 software. All the quantitative data are represented as the mean ± standard deviation (SD) in our experiment. Differences among multiple groups were determined *via* one-way Analysis of variance (ANOVA) with the SNK test. An independent sample $T$-test was used for heterogeneity of variance. Kruskal–Wallis H test was used for the measurement data that does not conform to normal distribution. Data with differences (probabilities of 0.05 or less for $P$-values) were regarded as statistically significant.

## RESULTS

### METTL3 was overexpressed in EOC

All port sediment cores collected were from an area that has a significant disturbance signature due to past dredging. We selected EOC ($n = 3$) and OE ($n = 3$) samples for transcriptome-wide detection of m6A modifications and got the top 100 differentially expressed genes, the differential expression heat map of which were shown in Fig. 1A. GO and KEGG analysis revealed that these genes were related to cell proliferation, extracellular matrix, and cancer- and cel-cycle-related pathways (Figs. 1B–1C). A heat map was created to study differential expression profiles of methylation-related genes in EOC and OE (Fig. 1D). Among these methylation-related genes, METTL3, ELF3, IGF2BP2, FTO, and METTL14 were verified to be overexpressed in EOC relative to the OE group, especially the METTL3 (Fig. 1E, $P < 0.05$).

### Silencing METTL3 suppressed the malignant phenotypes of EOC cells

We evaluated the efficiency of si-METTL3 and pcDNA3.1-METTL3 by qRT-PCR and western blot. The results unveiled that all three siRNA had inhibitory effects, among which si-METTL3#1 had the most significant inhibitory effect, so the si-METTL3#1 was used in subsequent experiments (Figs. 2A–2B, $P < 0.05$). At the same time, pcDNA3.1-METTL3 successfully promoted the mRNA and protein levels of METTL3 (Figs. 2A–2B, $P < 0.01$). Next, our findings uncovered that si-METTL3 or pcDNA3.1-METTL3 did not significantly affect the cell viability after 24 h of treatment, but after 48 h and 72 h of treatment, the former inhibited the cell viability, while the latter fostered the cell viability (Fig. 3C, $P < 0.05$). Furthermore, si-METTL3 attenuated the clonal expansion of EOC cells and induced the EOC cells to arrest in the GO/G1 phase, while pcDNA3.1-METTL3 did the opposite (Figs. 2D–2E, $P < 0.05$).

### Silencing METTL3 fostered the apoptosis of EOC cells by regulating FAS/FASL pathway

TUNEL assay exhibited that the apoptotic cells were increased by si-METTL3 and were diminished by pcDNA3.1-METTL3 (Fig. 3A, $P < 0.05$). To further analyze the molecular mechanism of METTL3 leading to the above results, we detected the expression of FAS/FASL pathway-related proteins. Si-METTL3 caused the upregulation of FAS, p-FADD/FADD, caspase-8, caspase-3, cleaved caspase-8, cleaved caspase-3, and Bax as well

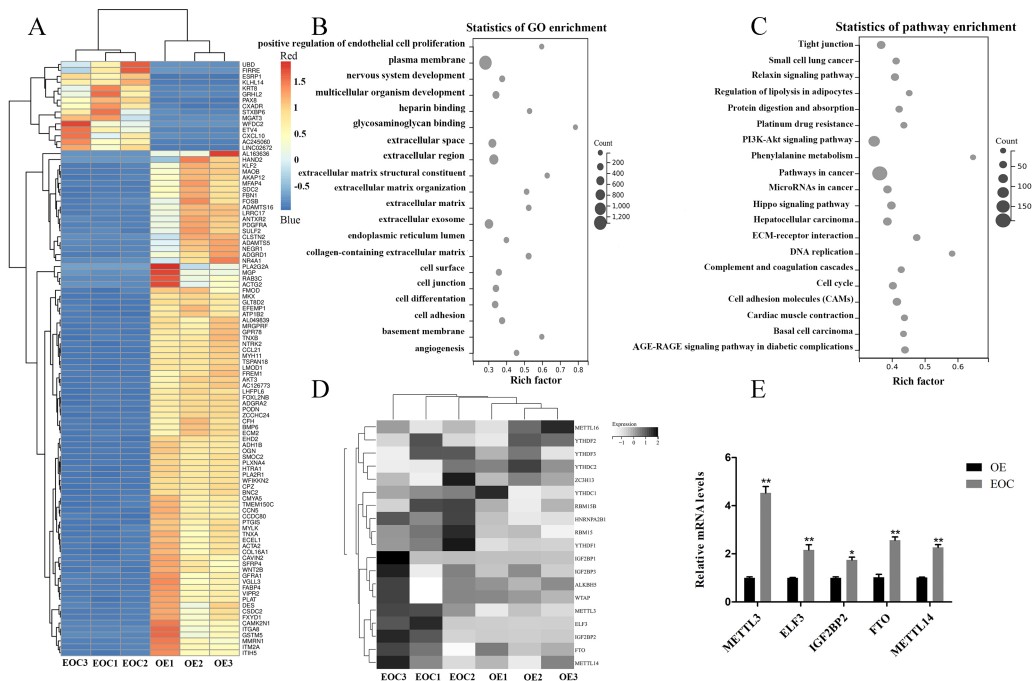

**Figure 1 Identification of methylation-associated genes in endometrioid ovarian cancer.** (A) Representative heatmap of differentially expressed genes between the endometrioid ovarian cancer (EOC) and ovarian endometriosis (OE). (B) Gene Ontology (GO) analysis of differentially expressed genes. (C) Kyoto Encyclopedia of Genes and Genomes (KEGG) analysis of differentially expressed genes. (D) Representative heatmap of methylation-associated genes. (E) Relative expression of methylation-associated genes METTL3, ELF3, IGF2BP2, FTO, and METTL14 in EOC and OE tissues. *$P < 0.05$, **$P < 0.01$ *vs.* OE group.

as the downregulation of Bcl-2, while pcDNA3.1-METTL3 did the opposite (Fig. 3B, $P < 0.05$). All these evidences indicated that silencing METTL3 fostered the apoptosis of EOC cells by regulating the FAS/FASL pathway.

## Sul dose-dependently inhibited viability promoted apoptosis and regulated the expression of IGF2BP2, FAS, KRT8, and METTL3

To screen for the working concentration of Sul, we treated EOC cells for different times (24, 36, and 48 h) with different concentrations (40, 60, 80, and 100 μM) of Sul and calculated different IC50 values (Fig. 4A), respectively. IC50 values for the three different treatment times were all around 60 μM. The results of flow cytometry illustrated that with the increase of Sul concentration from 20 μM to 60 μM, the apoptosis rate gradually elevated (Figs. 4B–4C, $P < 0.01$). Moreover, the mRNA contents of IGF2BP2 and FAS were upregulated, while those of KRT8 and METTL3 were downregulated as the Sul concentration raised (Fig. 4D, $P < 0.05$).

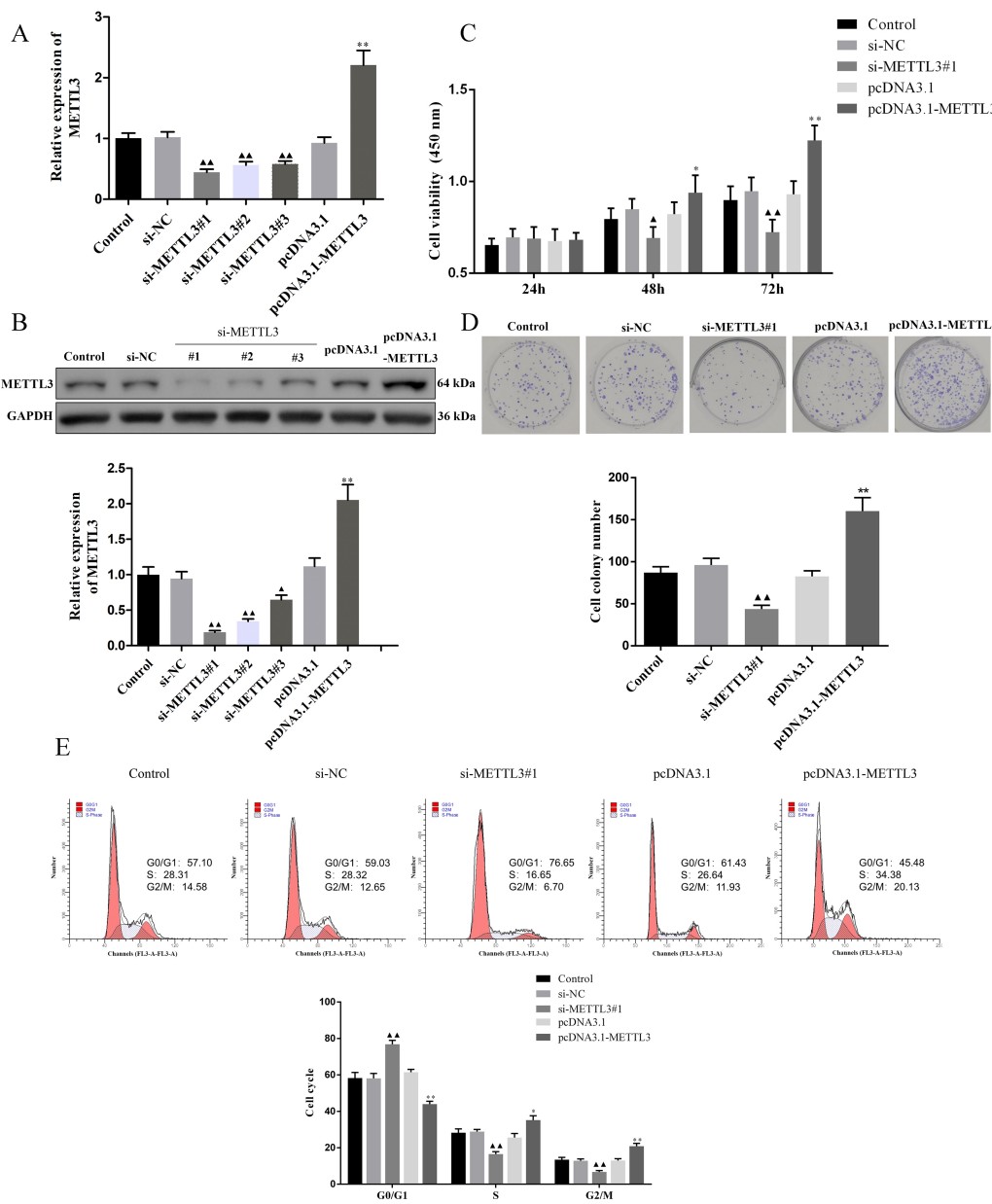

**Figure 2** **Effect of METTL3 on COV362 cell proliferation.** (A, B) Relative expression of METTL3 in COV362 cells transfected with siRNA against METTL3 or pcDNA3.1 overexpressed METTL3 was assessed by qPCR and Western blot analysis, respectively. (C) The viability of cells was evaluated by a CCK-8 assay. (D) Colony formation assays were performed in COV362 cells treated with si-NC, si-METTL3, pcDNA3.1, or pcDNA3.1-METTL3. (E) Representative images of the cell cycle in COV362 cells after silencing METTL3 or elevating METTL3 were analyzed by flow cytometry. ▲$P < 0.05$, ▲▲$P < 0.01$ *vs.* si-NC group; *$P < 0.05$, **$P < 0.01$ *vs.* PcDNA3.1 group.

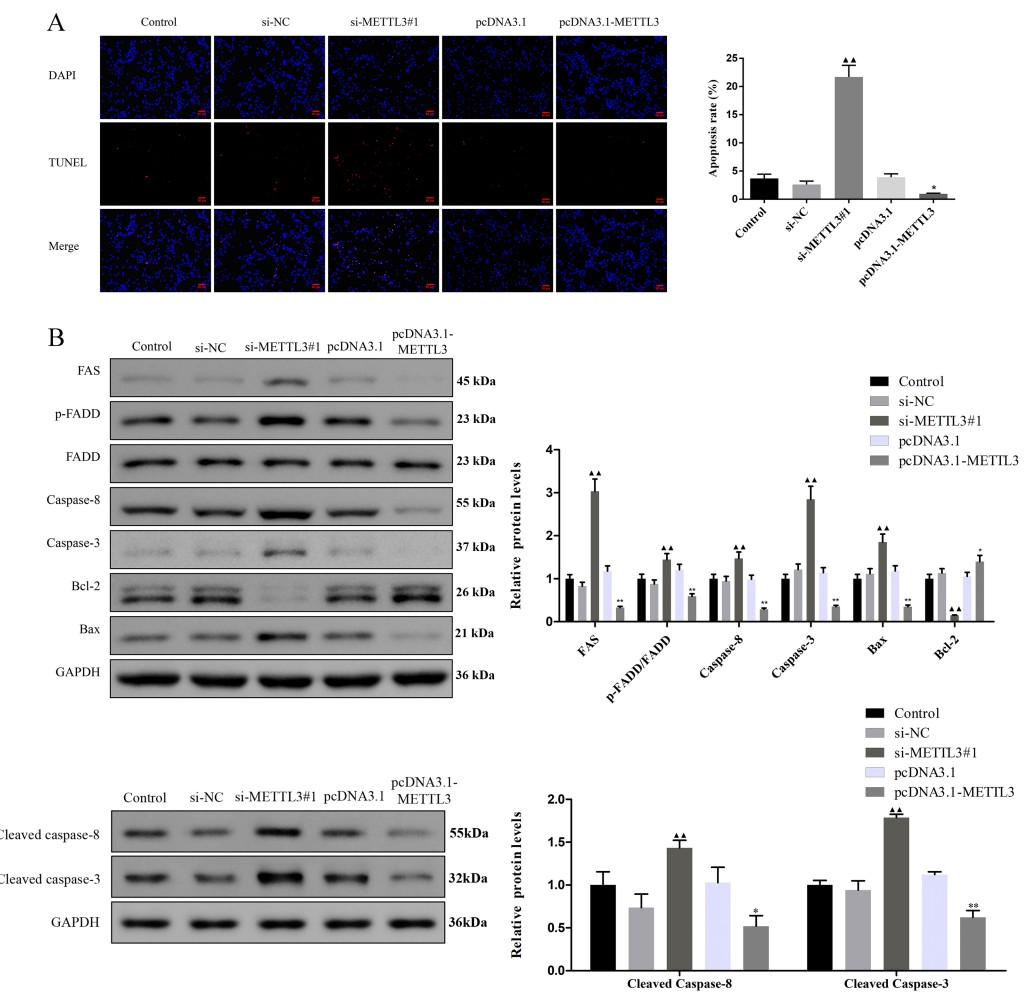

**Figure 3** **Effect of METTL3 on COV362 cell apoptosis.** (A) Representative images of cell apoptosis in COV362 cells after silencing METTL3 or elevating METTL3 were analyzed by Tunel staining. Scale bar =50 μm. (B) FAS, FADD, p-FADD, Caspase-8, Caspase-3, Bcl-2, and Bax protein expressions were determined in COV362 cells with METTL3 knock-down or overexpression through western blotting. ▲$P <$ 0.05, ▲▲$P < 0.01$ *vs.* si-NC group; *$P < 0.05$, **$P < 0.01$ *vs.* PcDNA3.1 group.

## Effect of pcDNA3.1-METTL3 on malignant phenotypes of EOC cells was reversed by Sul

To explore whether Sul has an anti-cancer effect by inhibiting METTL3, we treated the cells overexpressing METTL3 with Sul. Interestingly, the increases in cell viability (Fig. 5A, $P < 0.05$) and clonal expansion capacity (Fig. 5B, $P < 0.05$) as well as the decreases in apoptosis (Fig. 5C, $P < 0.05$) induced by pcDNA3.1-METTL3 in COV362 cells were reversed after Sul treatment (Figs. 5A–5C, $P < 0.01$). Furthermore, we found that the effect of METTL3 overexpression on the expression levels of FAS, FADD, p-FADD, Bcl-2, Bax, and cleaved Caspase-3 in COV362 cells was reversed under Sul treatment (Fig. 5D, $P < 0.01$). To sum up, these results showed that Sul might impede the malignant development of EOC by inhibiting METTL3.

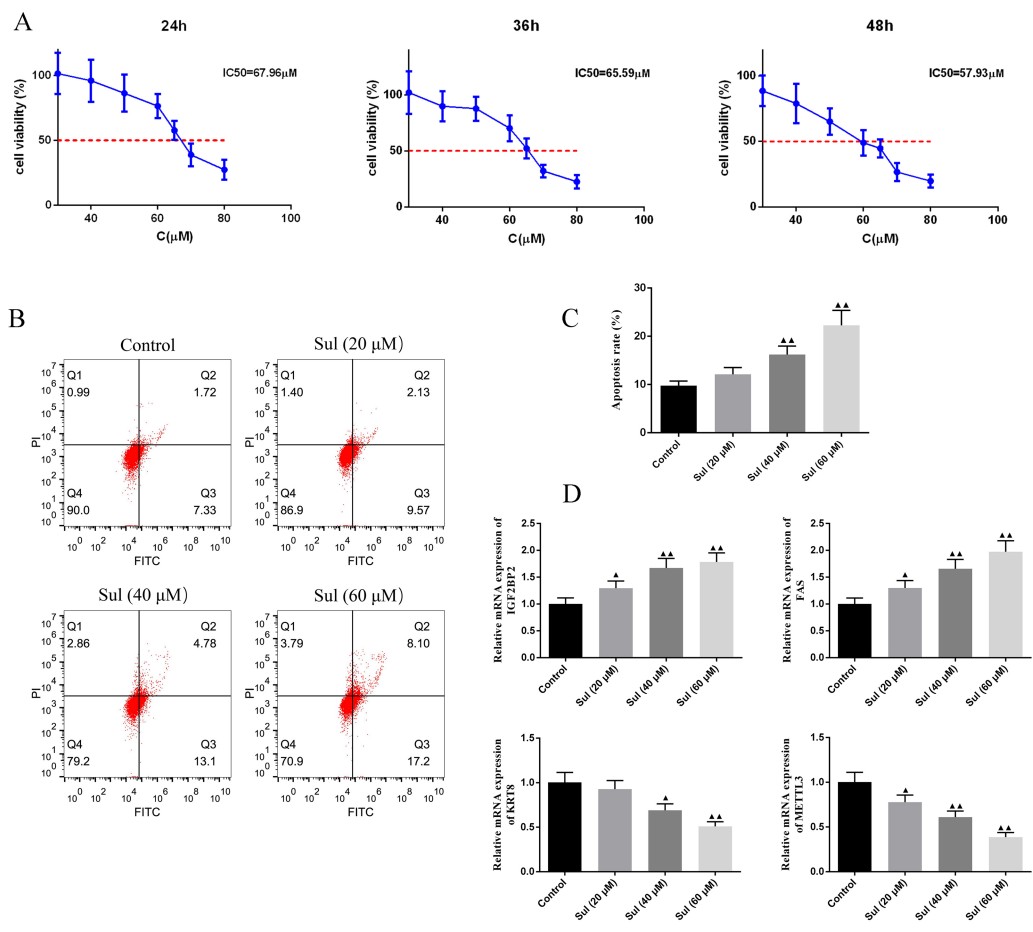

**Figure 4** **Sulforaphene promoted cell apoptosis of COV362 cells.** (A) CCK-8 analysis suggested that the cell viability of COV362 cells was inhibited by a dose-dependent manner of sulforaphene (Sul) treatment. (B) Representative pictures of apoptosis of Sul-treated COV362 cells that were measured by FACS analysis. (C) The relative apoptosis rate of COV362 cells treated with the different concentrations of Sul. (D) Effect of different concentrations of Sul on IGF2BP2, FAS, KRT8, and METTL3 mRNA levels, as assessed by qPCR. ▲$P < 0.05$, ▲▲$P < 0.01$ *vs.* control group.

## DISCUSSION

Accumulated evidence suggests that the dysmethylation of m6A is closely related to the development of OC and, in particular, affects the malignancy and prognosis of OC (*Zhang et al., 2021a*; *Zhang et al., 2021b*)). In this study, we determined that the METTL3 gene is the key gene that affects the development of EOC by combining MeRIP-seq and RNA sequencing. There has been evidence that METTL3 is highly expressed in human OC tissue and is related to lymph node metastasis and high pathological grade (*Liang et al., 2020*). Additionally, METTL3 is considered to be the primary gene responsible for OC metastasis, which can promote the maturation of miR-126-5p through m6A modification and activate the signal pathway related to survival, foster OC metastasis through m6A modification of TRPC3 mRNA, exert a carcinogenic effect on OC by stimulating AXL translation and

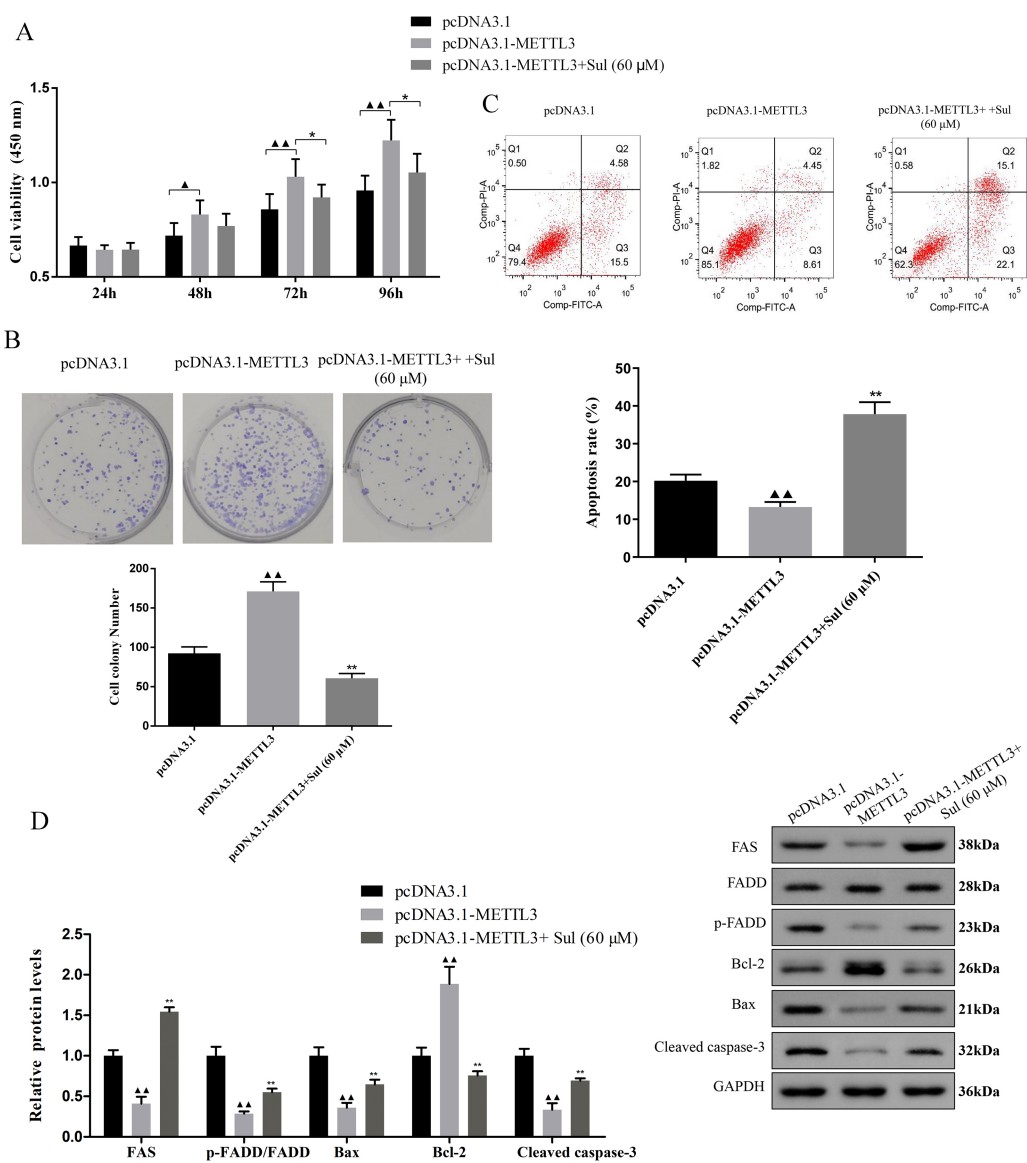

**Figure 5** **Sulforaphene prevented the effects of METTL3 overexpression on COV362 cells.** (A) Cell proliferation of COV362 cells was measured using CCK-8 assay at 24 h, 48 h, 72 h, and 96 h. (B) Cell proliferation was also assessed by clone formation assays in METTL3-overexpressed COV362 cells treated with high-dose sulforaphene. (C) Flow cytometry results show that the effects of sulforaphene increased cell apoptosis in the COV362 cells with METTL3 overexpression. $^{\blacktriangle}P < 0.05$, $^{\blacktriangle\blacktriangle}P < 0.01$ *vs.* pcDNA3.1 group; $^{*}P < 0.05$, $^{**}P < 0.01$ *vs.* pcDNA3.1-METTL3 group.

epithelial-mesenchymal transformation (*Hua et al., 2018*; *Shen et al., 2022*; *Bi et al., 2021a*; *Bi et al., 2021b*). Although the role of METTL3 in OC has been completely established, it is unknown if METTL3 functions in the same way throughout the various subtypes of OC. In our study, we mainly focused on EOC, an epithelial ovarian cancer, and found that silent METTL3 could inhibit cell viability and proliferative capacity, promote apoptosis, and induce cell cycle arrest in the G0/G1 phase, indicating that METTL3 performed a similar
role in EOC as it did in OC. In addition, *Ma et al. (2020)* pointed out that silencing METTL3 in another epidemiologic ovarian cancer can also weaken cell proliferation and migratory ability and promote cell apoptosis. These pieces of evidence indicate that METTL3 may act in a similar carcinogenic manner in various OC cell subtypes, which undoubtedly helps to improve the robustness of the preclinical research of METTL3 against OC.

We also found that silencing METTL3 can activate FAS/FADD apoptosis pathway and apoptosis-related pathway. The combination with FAS FASL can induce apoptosis of Fas-carrying cells, and this process requires the recruitment of FADD to produce active-caspase-8, which cleaves caspase-3 to initiate apoptosis (*Sun et al., 2022*). Activation of Fas/FADD is the key step to start the external apoptosis process, and the loss of related genes in the Fas/FADD apoptosis pathway will lead to a poor prognosis of OC (*Duiker et al., 2010*). In our investigation, the expression of FAS, p-FADD/FADD, caspase-8, and caspase-3 was promoted by silencing METTL3, which indicated that silencing METTL3 promoted the apoptosis of EOC cells by triggering FAS/FADD apoptosis pathway. At the same time, we unearthed that silencing METTL3 can also control the production of Bax and Bcl-2, which are the essential proteins in mitochondrial apoptosis. According to the accumulating data, increasing the ratio of Bax/Bcl-2 can effectively alleviate the metastasis and development of OC (*Zhang et al., 2019*; *Liu et al., 2017*). Previous research has reported that silencing METTL3 can increase the ratio of Bax/Bcl-2, trigger mitochondrial apoptosis, and prevent the development of lung cancer cells (*Wei, Huo & Shi, 2019*). These effects appear to be related to the inhibition of m6A modification of Bcl-2 by METTL3 (*Zhang et al., 2021a*; *Zhang et al., 2021b*). Similarly, our findings also suggested that silencing METTL3 may promote the apoptosis of EOC cells by triggering the apoptosis pathway by altering the Bax/Bcl-2 signaling pathway.

Although Sul has been mentioned in previous articles as an epigenetic regulator with the ability to regulate DNA methyltransferase, histone deacetylase, non-coding RNA, *etc* (*Su et al., 2018*), this study marks the first time that Sul has been proven to be capable of targeting m6A methyltransferase METTL3. In addition, although the role of Sul in OC has been extensively covered, its role in EOC has only recently come to light. Consistent with the anticancer effect of Sul in OC (*Kan, Wang & Sun, 2018*), we found that Sul fostered EOC cell apoptosis and diminished EOC cell proliferative capacity in a concentration-dependent manner. Sul reversed the carcinogenesis of pcDNA3.1-METTL3, which indicated that Sul partly prevented the development of EOC by reducing the expression of METTL3. Furthermore, Sul upregulated the levels of insulin growth factor 2 mRNA binding protein 2 (IGF2BP2) and FAS as well as downregulated the levels of KRT8 and METTL3. IGF2BP2, localized on chromosome 3q27, a m6A-modified reading protein, plays an important role in embryonic development that is lowly expressed in normal adult tissues and can act as a post-transcriptional regulator of mRNA localization, stabilization, and translation by stabilizing mRNAs to extend their half-life, while KRT8 is a marker of epithelium, both of which are reported to be related to the development of OC (*Prayudi et al., 2020*; *Park et al., 2022*). The link between METTL3 and KRT8 is yet unknown, however, it appears that IGF2BP2 is necessary for METTL3's regulatory action on m6A modification (*Wang et al., 2022*). IGF2BP2 is up-regulated in a variety of tumors, and its biological activity

contributes to cancer formation by interacting with various non-coding RNAs including microRNAs (miRNAs), long non-coding RNAs (lncRNAs), and circular RNAs (circRNAs) (*Wang, Chen & Qiang, 2021*). It was surprising that the expression of IGF2BP2 increased in COV362 cells treated with Sul. As a result, there may be a part of the intracellular compensatory response in EOC. More importantly, our study also has some limitations. Future studies are necessary to address whether a specific pathway or mechanism by which Sul affects IGF2BP2 directly. And, the more details of clinical trials and basic scientific research need further study in the future.

In conclusion, our study shows that the knockdown of METTL3 has significantly inhibited cell proliferation and promoted cell apoptosis *in vitro*, while an opposite effect was observed in COV362 cells with METTL3 overexpression. Additionally, we report that the Sul contributed to an increase in cell apoptosis of COV362 cells, and disrupted the effect of METTL3 overexpression on cell proliferation and apoptosis in EOC. These results showed that Sul-induced METTL3-dependent apoptosis, which might be related to the FAS/FADD and Bax/Bcl-2 associated pathway, might provide its clinical potential applications for preventing and treating EOC.

### Funding
This research was funded by the Public Welfare Technology Project of Zhejiang Province (LGF19H160011) and the Zhejiang Provincial Health and Medicine Science and Technology Project (2019ZD001). The funders had no role in study design, data collection and analysis, decision to publish, or preparation of the manuscript.

### Grant Disclosures
The following grant information was disclosed by the authors:
Public Welfare Technology Project of Zhejiang Province: LGF19H160011.
Zhejiang Provincial Health and Medicine Science and Technology Project: 2019ZD001.

### Competing Interests
The authors declare there are no competing interests.

### Author Contributions

- Hui-Yan Yu conceived and designed the experiments, performed the experiments, analyzed the data, prepared figures and/or tables, authored or reviewed drafts of the article, and approved the final draft.
- Li Yang performed the experiments, prepared figures and/or tables, and approved the final draft.
- Yuan-Cai Liu performed the experiments, prepared figures and/or tables, and approved the final draft.
- Ai-Jun Yu conceived and designed the experiments, authored or reviewed drafts of the article, and approved the final draft.

## Human Ethics

The following information was supplied relating to ethical approvals (i.e., approving body and any reference numbers):

The studies involving human participants were reviewed and approved by The Ethics Committee of Zhejiang Cancer Hospital (No. ZJSZLYY-2021-05-146). The patients/participants provided their written informed consent to participate in this study.

## Data Availability

The raw measurements are available in the Supplementary Files.

## Supplemental Information

Supplemental information for this article can be found online at http://dx.doi.org/10.7717/peerj.16308#supplemental-information.

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
