# Peer review of "Sulforaphene suppressed cell proliferation and promoted apoptosis of COV362 cells in endometrioid ovarian cancer"

_PeerJ, doi:10.7717/peerj.16308_

## Round 0.1 · original submission · Minor Revisions

Please respond and make appropriate revisions based on the Reviewers' suggestions and my comments (below). This will greatly improve the quality of this manuscript.

My comments:
The dysregulated m6A RNA methylation genes in ovarian cancer were found in this study by the use of MeRIP-seq and RNA sequencing. Additionally, the study investigated the effects of sulforaphane on METTL3. The findings of the study indicate that METTL3 plays a crucial role in the survival and proliferative capacity of ovarian cancer cells. Additionally, it was shown that sulforaphane has the potential to decrease cell proliferation and cause apoptosis by suppressing the activity of METTL3.

However, issues that need to be revised were still detected:
1. There's an inconsistency in the naming of "Sulforaphane". It's referred to as "Sul," while later it's spelled out as "Sulforaphene"? Please confirm this inconsistency, carefully check the remaining part of this paper (including line 1: Title), and make appropriate revisions.
2. The authors should provide a detailed explanation for the observed up-regulation of IGF2BP2, an oncogene, in the context of sulforaphane's apparent anti-cancer effects. If the up-regulation of IGF2BP2 is unexpected given its oncogenic role, the authors should consider proposing hypotheses to explain this finding. For example, could there be a specific pathway or mechanism by which sulforaphane affects IGF2BP2 differently in this context? Or might this up-regulation be part of a compensatory response within ovarian cancer cells?
3. "(Chatterjee et a., 2016)". It likely should be "(Chatterjee et al., 2016)".
4. [resulting in the decrease of OC's apoptosis rate] should be [resulting in a decrease in the apoptosis rate of OC].
5. The text "Clinical specimen86" appears to have a formatting issue.
6. Please explain why the human EOC cell line COV362 was selected in this study, considering that a large number of ovarian cancer cell models have been reported.
7. [knockout of METTL3] should be [knockdown of METTL3].
8. [Adding paraformaldehyde solution to each chamber slide, the cells were fixed for 20 minutes] should be revised to [The cells were fixed for 20 minutes by adding paraformaldehyde solution to each chamber slide].
9. The use of "si-METTL3" and "oe-METTL3" may lead to confusion. It might be clearer to define these terms as specific treatments, such as "siRNA targeting METTL3" and "overexpression vector for METTL3" or similar terms, depending on the context.
10. It would be beneficial to define EOC and OE at their first mention for the reader's understanding. While EOC likely refers to epithelial ovarian cancer, OE is not defined and may confuse readers.
11. [yet that of KRT8 and METTL3 were downregulated] should be [the mRNA contents of IGF2BP2 and FAS were upregulated, while those of KRT8 and METTL3 were downregulated].
12. Line 286: There's a typo in the text "MTL3" which should be corrected to "METTL3."
13. The Conclusion might benefit from a concise summary that highlights the key findings and implications of the study. While the current conclusion does this to some extent, a more focused summary could enhance the impact of the text.
14. [These evidences? aloneg?]: Careful proofreading and language editing is required to improve the quality of this study.

**Language Note:** The Academic Editor has identified that the English language must be improved. PeerJ can provide language editing services - please contact us at copyediting@peerj.com for pricing (be sure to provide your manuscript number and title). Alternatively, you should make your own arrangements to improve the language quality and provide details in your response letter. – PeerJ Staff

·

Basic reporting

Endometrioid ovarian cancer (EOC) is one rare kind of ovarian cancer (OC), whose development is related to endometriosis. At this stage, whether M6A plays roles in EOS is still unknown. Yu HY and colleagues presented a topic entitled “Sulforaphene suppressed cell proliferation and promoted apoptosis of COV362 cells in endometrioid ovarian cancer by inhibiting m6A methyltransferase METTL3”. Overall, this topic is very interesting and the data presented are convincing. However, there are some concerns that need to be addressed, which are listed as follows:

1. The resolution of Figures are poor, such as Figure 1. Please change these figures with high quality Figures.

2. The authors should provide the molecular weight for all WB blots.

3. For the apoptosis signaling pathways, the authors should also check the cleavage of caspases such as cleaved-cas8 and cleaved-cas3.

4. Some grammar errors have been identified. The authors should check the whole manuscript carefully.

Experimental design

The experimental design is ok.

Validity of the findings

This topic is very interesting and the data presented are convincing.

·

Basic reporting

There are still some issues that need to be addressed.

Experimental design

1. There were only three EOC samples and three OE samples, which may lead to a lack of confidence in the results.
2. EOC is rare, as an independent histological subtype, it is still difficult to diagnose clinically and is often confused with other types, such as high-grade serous ovarian carcinoma, endometrioid carcinoma and mixed epithelial carcinoma. In particular, grade 3 EOC may mimic high-grade serous ovarian carcinoma. Therefore, it is recommended to supplement the identification of EOC samples.

Validity of the findings

1. Among the methylation-related genes, METTL3, ELF3, FTO and METTL14 were significantly highly expressed in ovarian cancer tissues. Whether METTL3 is the core gene of methylation-related genes can be identified according to METTL3 changes most significantly. Moreover, the number of samples tested was extremely limited.

2.The authors conclude that Sul could suppress cell proliferation and promote apoptosis of EOC cells by inhibiting the METTL3 to activate the FAS/FADD and mitochondrial apoptosis pathways. However, the authors only examined the effect of METTL3 intervention on FAS/FADD and did not examine the effect of Sul on FAS/FADD after METTL3 intervention.

Additional comments

NONE

---

## Round 0.2 · Minor Revisions

Issues that need to be further improved:

1. Abstract: [Silencing METTL3 attenuated … METTL3 overexpression on EOC cells]: These sentences are quite long and contain a lot of information, which can make it difficult for the readers to follow. Breaking it up into smaller sentences or using commas to separate different ideas can help improve readability.

Here's an example: "Silencing METTL3 reduced the clonal expansion and viability of EOC cells, and caused the cells to arrest in the G0/G1 phase. This also promoted apoptosis in the EOC cells and activated the FAS/FADD and mitochondrial apoptosis pathways. In contrast, overexpressing METTL3 had the opposite effect. Sul, in a dose-dependent manner, reduced the viability of EOC cells but promoted their apoptosis. Sul also increased the levels of IGF2BP2 and FAS, while decreasing the levels of KRT8 and METTL3. Furthermore, Sul was able to reverse the effects of METTL3 overexpression on EOC cells."

2. The authors did not examine whether, in Sul-treated cells, overexpression of METTL3 could promote cell viability that was suppressed by Sul. Therefore, to be safe, they may have two options: the first is to modify the Title of this paper by deleting [by inhibiting m6A methyltransferase METTL3]. The second is to add the results that support the ability of METTL3 overexpression to rescue cell viability in EOC cells treated with Sul.

3. Line 238: Please delete [and METTL3 was down-regulated], because they have written this [KRT8 and METTL3 were downregulated].

·

Basic reporting

The authors have well addressed my concerns.

Experimental design

The authors have well addressed my concerns.

Validity of the findings

The authors have well addressed my concerns.

·

Basic reporting

This study investigated that sulforaphane (Sul) is implicated in EOC development by regulating methyltransferase-like 3 (METTL3). The author has revised it according to my suggestions.

Experimental design

The revised article meets the requirements for publication.

Validity of the findings

no commen

Additional comments

The author needs to check the language details in further.

---

## Round 0.3 · accepted · Accept

My concerns have been well addressed and I think this revised article can be considered for publication in this journal.